# Effect of a Screening and Education Programme on Knowledge, Beliefs, and Practices Regarding Osteoporosis among Malaysians

**DOI:** 10.3390/ijerph19106072

**Published:** 2022-05-17

**Authors:** Chin Yi Chan, Shaanthana Subramaniam, Kok-Yong Chin, Soelaiman Ima-Nirwana, Norliza Muhammad, Ahmad Fairus, Pei Yuen Ng, Jamil Nor Aini, Noorazah Abd Aziz, Norazlina Mohamed

**Affiliations:** 1Department of Pharmacology, Universiti Kebangsaan Malaysia Medical Centre, Cheras 56000, Malaysia; chanchinyi94@gmail.com (C.Y.C.); shaanthana_bks@hotmail.com (S.S.); chinkokyong@ppukm.ukm.edu.my (K.-Y.C.); imasoel@ppukm.ukm.edu.my (S.I.-N.); norliza_ssp@ppukm.ukm.edu.my (N.M.); 2Department of Anatomy, Universiti Kebangsaan Malaysia Medical Centre, Cheras 56000, Malaysia; fairusahmad@ukm.edu.my; 3Faculty of Pharmacy, Universiti Kebangsaan Malaysia Kuala Lumpur Campus, Jalan Raja Muda Abdul Aziz, Kuala Lumpur 50300, Malaysia; pyng@ukm.edu.my; 4Faculty of Health Science, Universiti Kebangsaan Malaysia Kuala Lumpur Campus, Jalan Raja Muda Abdul Aziz, Kuala Lumpur 50300, Malaysia; ainijamil@ukm.edu.my; 5Department of Family Medicine, Universiti Kebangsaan Malaysia Medical Centre, Cheras 56000, Malaysia; azah@ppukm.ukm.edu.my

**Keywords:** awareness, attitude, bone, behaviour, calcium, education, lifestyle

## Abstract

Background: Osteoporosis is an emerging geriatric condition with high morbidity and healthcare cost in developing nations experiencing rapid population ageing. Thus, identifying strategies to prevent osteoporosis is critical in safeguarding skeletal health. This study aimed to evaluate the effects of a bone health screening and education programme on knowledge, beliefs, and practice regarding osteoporosis among Malaysians aged 40 years and above. Methods: A longitudinal study was conducted from April 2018 to August 2019. During the first phase of the study, 400 Malaysians (190 men, 210 women) aged ≥ 40 years were recruited in Klang Valley, Malaysia. Information on subjects’ demography, medical history, knowledge, and beliefs regarding osteoporosis, physical activity status, and dietary and lifestyle practices were obtained. Subjects also underwent body anthropometry measurement and bone mineral density scan (hip and lumbar spine) using a dual-energy X-ray absorptiometry device. Six months after the first screening, similar investigations were carried out on the subjects. Results: During the follow-up session, 72 subjects were lost to follow up. Most of them were younger subjects with a lower awareness of healthy practices. A significant increase in knowledge, beliefs (*p* < 0.05), calcium supplement intake (*p* < 0.001), and dietary calcium intake (*p* = 0.036) and a reduction in coffee intake (*p* < 0.001) were found among subjects who attended the follow-up. In this study, the percentage of successful referrals was 41.86%. Subjects with osteoporosis were mostly prescribed alendronate plus vitamin D3 by medical doctors, and they followed the prescribed treatment accordingly. Conclusions: The bone health screening and education programmes in this study are effective in changing knowledge, beliefs, and practice regarding osteoporosis. The information is pertinent to policymakers in planning strategies to prevent osteoporosis and its associated problems among the middle-aged and elderly population in Malaysia. Nevertheless, a more comprehensive bone health education program that includes long-term monitoring and consultation is needed to halt the progression of bone loss.

## 1. Introduction

Osteoporosis is a geriatric condition characterised by bone mass and microarchitecture deterioration. This often neglected disease carries high morbidity and healthcare cost in developing nations experiencing rapid population ageing [1,2]. For instance, hip fracture incidence in Malaysia is projected to increase from 5880 cases in 2018 to 20,893 cases by 2050, representing the highest increase in Asia [3]. In addition, dual-energy X-ray absorptiometry (DXA) machines are reserved for osteoporosis diagnosis for high-risk patients and monitoring treatment progress instead of public screening in these countries [4]. Hence, many people are unaware of their bone health status. Therefore, identifying strategies to prevent osteoporosis is critical to safeguarding skeletal health.

Osteoporosis is a preventable disease. Apart from non-modifiable factors such as age-associated biochemical changes in the body and genetics, modifiable factors such as lifestyles, physical activity, and dietary factors also influence the development of osteoporosis [5]. Current evidence supports the effectiveness of public education on osteoprotective behaviours, such as physical activity and proper nutrition, in preventing bone loss [6]. Other key components of osteoporosis prevention include improving public knowledge about osteoporosis, modifying public attitudes towards preventive behaviours, and motivating the public to undertake preventive actions and make them a routine [7,8]. 

Various types of educational programs have been attempted to enhance knowledge, beliefs, or practice regarding osteoporosis. Intervention methods used include weekly educational programmes [9,10,11,12], lectures with group discussions [13], and educational leaflets [6]. The outcomes of these interventions are quite promising, but the extent to which adults retain the knowledge, beliefs, and healthy lifestyles requires further investigation. A recent systematic review reported limited evidence for the effectiveness of patient education on osteoporosis, based on improvements in physical discomfort, disability, health-related quality of life, adherence and persistence, and knowledge [14]. The efficacy of osteoporosis screening and educational programmes in Malaysia has not been reported.

Thus, this study aims to determine the effects of a screening and education programme on knowledge, beliefs, and practice regarding osteoporosis among Malaysians aged 40 years and above. It is hypothesised that this programme would improve the knowledge and beliefs regarding osteoporosis among the subjects and prompt them to modify their behaviour for the betterment of their bone health.

## 2. Materials and Methods

This is a longitudinal study involving Malaysians aged 40 years and above in Klang Valley (Kuala Lumpur and its environs), Malaysia. Subject recruitment and screening have been described in previous publications [15,16,17,18,19]. Briefly, subjects were recruited via quota sampling based on sex (1:1) and ethnicity (Malay 45%, Chinese 45%, Indian 10%). The proportion was similar to the demographic characteristics of Kuala Lumpur [20]. Invitations with specific inclusion and exclusion criteria were sent to community centres in Klang Valley. Recruitment was also advertised through local newspapers and radio broadcasts. Potential participants were interviewed over the phone to ensure their eligibility. Only subjects fulfilling the inclusion criteria were recruited. Subjects with mobility problems, taking medications (glucocorticoids, sex hormone replacement, sex-hormone deprivation agents, thyroid supplements, thiazide diuretics, anticonvulsants, anti-osteoporosis drugs excluding calcium and vitamin D), or having medical conditions affecting bone health (hyper/hypocalcaemia, hyper/hypoparathyroidism, rickets, osteomalacia, Paget’s disease, chronic renal diseases) were excluded from this study. Subjects who had a fracture two years before the screening date were excluded because their bone health status and lifestyle might be different from subjects without fractures. Those having metal implants at the scanning sites were also excluded. 

The study protocol was reviewed and approved by the Ethics Committee of Universiti Kebangsaan Malaysia Medical Centre (approval code: UKM PPI/111/8/JEP 2017-721). The subjects provided informed consent before participating in this study. During the first phase of the study, subjects’ demographic details, medical history, knowledge, beliefs, and dietary and lifestyle practices regarding osteoporosis were collected via a questionnaire. The subjects completed the questionnaire at the study centres themselves. Age was determined from the subjects’ identification cards. Ethnicity, sex, menstrual status, age of menarche, age of menopause, parity, and presence of pre-existing medical conditions and medical treatments were self-declared. Subjects’ occupations were categorised as manual or sedentary based on the amount of time they spent walking or carrying heavy objects or sitting at the workplace or in a vehicle. The subjects were classified based on household income into the bottom 40% (B40, with monthly household income < RM 7640), the middle 40% (M40, with monthly household income RM 7640–15,159) and the top 20% (T20, with monthly household income > RM 15,160) groups according to data from the Malaysian census [21].

Subjects’ knowledge and health beliefs regarding osteoporosis were collected using a modified Osteoporosis Prevention and Awareness Tool (OPAAT) [22] and Osteoporosis Health Beliefs Scale (OHBS) [23], respectively. The details of the questionnaire have been explained previously [17,18]. Briefly, OPAAT consists of a list of statements about osteoporosis that the subjects would rate true/false/do not know. Each correct answer would give one mark; otherwise, zero marks were awarded. OHBS consists of a list of statements of subjects’ beliefs on osteoporosis. The subjects would rate each statement using the Likert scale (1/strongly disagree to 5/strongly agree). A higher OPAAT score indicates a better knowledge level, while a higher OHBS score indicates a more positive attitude towards bone health. In terms of practice, the subjects disclosed their smoking behaviour, intake of dairy products (milk, yoghurt and cheese), beverages (coffee, tea and alcohol beer, wine or spirits), and use of calcium supplements [24]. The categorisation of subjects’ diet and lifestyle practices has been described previously [15,16,17,18,19]. The dietary intake of subjects was collected by using a diet history questionnaire [25], wherein subjects recalled the average dietary intake for the past 7 days. Data collected were analyzed by using Nutritionist Pro Software (Axxya Systems LLC, Woodinville, WA, USA). The physical activity status of the subjects was determined using the International Physical Activity Questionnaire (IPAQ), which is available online and free for use [26]. Briefly, subjects were required to recall the average amount of time spent in high-intensity activity, moderate-intensity activity, walking, and sitting/lying down (except sleeping) in a week. Subjects were classified into inactive, minimally active, or HEPA (health-enhancing physical activity) based on the total MET score or other additional criteria [27].

Upon completion of the questionnaires, anthropometric measurements were performed on the subjects. The standing height of the subjects without shoes was measured using a stadiometer (Seca, Hamburg, Germany) and recorded to the nearest 1 cm. The body weight of the subjects with light clothing and without shoes was determined using a weighing scale (Tanita, Tokyo, Japan) and was recorded to the nearest 0.1 kg. The body mass index (BMI) of the subjects was calculated by dividing the weight in kg by the squared height in meters. Generally, for subjects < 65 years, BMI < 18.5 kg/m^2^ was classified as underweight, 18.5–24.9 kg/m^2^ as normal, 25.0–29.9 kg/m^2^ as overweight, and >30.0 kg/m^2^ as obese [28]. For subjects ≥ 65 years old, a BMI of <22kg/m^2^ was underweight, 22–27 kg/m^2^ was normal, and >27 kg/m^2^ was overweight [29]. The waist circumference of the subjects, measured at the midpoint between the lowest rib margin and the iliac crest while subjects maintained a standing position using a soft measuring tape, was recorded to the nearest 0.1 cm. 

The bone mineral density of the subjects at the lumbar spine and femur of the non-dominant leg (femoral neck and total hip) was measured with DXA (Discovery QDR Wi, Hologic, MA, USA) by a single trained technician throughout the study period. The machine was calibrated daily using a phantom as per the manufacturer’s instructions. The short-term in-vivo coefficient of variation for the DXA machine was 1.8% and 1.2% for the lumbar spine and total hip, respectively [30]. The body fat percentage, lean body mass, lumbar spine BMD (average of L1–L4), and hip BMD were computed automatically by the DXA scanner. The T-score was generated by comparing the BMD values of the subjects with the sex and ethnic-specific reference values of the Asian population. According to the WHO guidelines, a T-score of ≤−2.5 indicates osteoporosis, between −2.5 and −1 indicates osteopenia, and >−1 indicates normal bone health status [31].

After the health screening process, subjects were consulted about their health status by physicians at the screening site. The physicians reviewed and explained the health screening results to the subjects. They also addressed the concerns about bone health status raised by the subjects. The subjects were consulted about ways to improve bone health through diet and physical activity with the aid of a booklet and atlas of calcium-rich food. The booklet contained basic information about osteoporosis, risk factors, osteoporosis-preventive steps that can be adopted by subjects, and contact details of the research team. Apart from that, subjects with osteoporosis were referred to the Primary Care Clinic, Universiti Kebangsaan Malaysia, or other health facilities preferred by the subjects. All subjects were given the booklet and DXA report, while those diagnosed to have osteoporosis also received a referral letter to the health facilities of their choice. Subjects were reminded of their follow-up after 6 months.

Subjects were followed up 6 months after the first screening. Similar investigations were carried out as per the first screening. Besides, subjects were asked if they had taken any steps to modify their diet or physical activity or had met any medical doctor to discuss their health screening results. For those subjects with osteoporosis who previously received a referral letter, they were asked whether they went to see a medical doctor, what treatments were received, and their compliance with the treatment.

### Statistical Analysis

Normality of the data was determined using the Kolmogorov–Smirnov Test. Skewed data were transformed logarithmically for analysis. Basic characteristics of the subjects and their scores for knowledge, beliefs, and practices regarding osteoporosis were expressed as the mean ± standard deviation for continuous data and as a percentage for categorical data. The comparison of characteristics between men and women, and between subjects who attended and lost to follow-up, was performed using the independent t-test for continuous variables, or the Chi-square test for categorical variables. Changes in subjects’ characteristics, BMD, knowledge, beliefs, or practice regarding osteoporosis between the baseline and follow-up were compared using the paired t-test for continuous variables, or the McNemar’s test for categorical variables. Written feedback on barriers to the adoption of osteoprotective behaviour and reasons for refusal to meet medical doctors was collected, and thematic analysis was carried out. A *p*-value of <0.05 was considered statistically significant. All statistical analyses were performed using SPSS Version 23 (IBM Corporation, Armonk, NY, USA). 

## 3. Results

### 3.1. Characteristics of Subjects during the Recruitment

A total of 400 subjects (47.5% men and 52.5% women) were recruited in the first phase of the study. The average age of men and women subjects was 57.78 ± 9.58 and 56.07 ± 8.10 years, respectively. The distribution of participants by ethnicity was Chinese 48.3%, Malay 42.3% and Indians and others 9.5%. Most of the subjects were married (92.8%), sedentary (93.0%), and had an estimated monthly salary of less than RM 7640 (94.8%). Most of them had at least a secondary school education (49.3%) and a normal BMI (45.8%). Among the women, the average age for menarche was 13.05 ± 1.87 years old. Most of them (49.5%) had 1 to 3 pregnancies in their lifetime. Most of the women in the study were postmenopausal (69.5%) with the average age of menopause being 51.08 ± 3.59 years old, and the average years since menopause 9.01 ± 5.98 years. Table 1 shows the baseline characteristics of the subjects.

### 3.2. Comparison of Characteristics of Subjects Compliant with or Lost to Follow Up 

A total of 72 study subjects did not attend the follow-up. Those who did not attend the follow-up session were generally younger (*p* = 0.002), had higher weight (*p* = 0.002), BMI (*p* < 0.001), waist circumference (*p* = 0.002), lower hip (*p* = 0.036) and spine T-score (*p* = 0.048), lower hip BMD (*p* = 0.041), and higher carbohydrate intake (*p* = 0.017) compared to subjects attending the follow-up (Table 2).

### 3.3. Characteristics of Subjects before and after Intervention

In general, most characteristics of the 328 participants did not differ significantly between baseline and follow-up apart from a few exceptions (Table 3). The hip T-score of the study participants at follow-up was significantly reduced compared to the baseline (*p* < 0.001). The scores for basic knowledge (*p* < 0.001) and prevention of osteoporosis (*p* < 0.001) increased during the follow-up. For health beliefs regarding osteoporosis, perception of susceptibility to osteoporosis decreased during the follow-up (*p* < 0.001). On the other hand, perception of the seriousness of osteoporosis (*p* < 0.001) and beliefs in the benefits of exercising (*p* = 0.002) increased during the follow-up. The distribution of responses to osteoporosis knowledge and beliefs questions is reported in Appendix A. 

This study also found that calcium supplement intake habits increased (*p* = 0.001) while coffee and tea intake habits decreased during follow-up (*p* = 0.001). For dietary intake, calcium (*p* = 0.018) and copper (*p* = 0.042) intakes were increased, while α-tocopherol (*p* = 0.028) and selenium (*p* = 0.024) intakes decreased during the follow-up (Table 3).

### 3.4. Barriers to Achieve Optimal Bone Health through Osteoprotective Practices

Most of the subjects attending the follow-up did not change their practices related to bone health, and their written responses were analysed. Most subjects did not take up calcium supplements because they did not feel the need for supplements (*n* = 138/257) and preferred to get calcium through food (*n* = 114/257). Other reasons for not taking calcium supplements were the high cost of the supplements (*n* = 2/257) and fear of constipation (*n* = 2/257) and gallstones (*n* = 1/257). Subjects were hesitant to take up dairy products because they did not habitually consume these products (*n* = 146/202) and did not like the taste of milk (*n* = 42/202). Other concerns included high fat content (*n* = 6/202), high cost of dairy products (*n* = 4/202), and lactose intolerance (*n* = 4/202). Most of them did not exercise because they were busy with work or house chores (*n* = 137/142), having health issues such as knee pain or leg oedema (*n* = 4/142) or lacking a companion (*n* = 1/142) (Appendix A).

### 3.5. Changes in Hip and Spine BMD after Intervention

Overall, only hip BMD decreased significantly during the follow-up compared to the baseline (*p* < 0.001) (Table 4). The changes in the hip and spine BMD values were close to the CV of the machine. Therefore, these changes could be due to random errors. Sub-analysis based on sex revealed that hip BMD decreased significantly in men, while hip and spine BMD decreased significantly in women regardless of menstrual status (*p* < 0.001). 

### 3.6. Referral Information and % of Successful Referrals and Reasons for Not Meeting Doctors

In this study, 43/49 of the new cases of osteoporosis attended the follow-up. Based on the 43 subjects, the percentage of successful referrals was 41.86% (*n* = 18/43). A total of 37.21% (*n* = 16/43) of subjects asked for referral letters but did not visit the doctors. They explained that they were busy with work/religious class (43.75%, *n* = 7/16), more comfortable with lifestyle changes at home (37.50%, *n* = 6/16), and several of them were afraid to take medications (18.75%, *n* = 3/16) (Appendix A). Examples of the written response given by subjects not meeting their doctors for referral are presented in Table 5.

### 3.7. Treatment Prescribed and Compliance of Subjects Who Met Medical Doctors with Referral Letters

All subjects referred to the medical doctors were prescribed pharmacological agents or lifestyle changes. Most of them were given alendronate plus vitamin D3 (44.44%, *n* = 8/18), followed by calcium supplements (33.33%, *n* = 6/18). Meanwhile, 27.78% (*n* = 5/18) of participants did not receive any medication but were advised to perform lifestyle changes. All of them followed a given prescription (Appendix A).

## 4. Discussion

Osteoporosis screening and education programmes have been conducted by the health authorities and non-governmental organisations in Malaysia to promote bone health. However, no studies have been performed to determine whether such programmes are effective in improving the knowledge, perceptions, and osteoprotective practices of the public. This study showed that the osteoporosis screening and education programme improved the knowledge and attitude of the subjects regarding osteoporosis. However, their perceived susceptibility towards osteoporosis decreased during the follow-up. The supplementary and dietary calcium intake and dairy product consumption increased significantly, while coffee and tea drinking reduced during follow-up. Forty-three new cases of osteoporosis were found in the screening, but only 41.86% of the patients visited doctors with the referral letters given. Patients who visited the doctors were prescribed alendronate plus vitamin D3, calcium supplements, or lifestyle changes. 

During the follow-up, 72 subjects (37 men and 35 women) did not attend the follow-up phase. Subjects lost to follow-up were younger and had low awareness of bone health. We did not examine the reasons hindering these subjects from attending a health screening. We postulate that, due to their young age, they were less concerned about their bone health. Moreover, they might not regard bone health as an important thing in life, and they may also be busy with work. A previous qualitative study among younger men in Malaysia showed that they had low risk perception towards diseases, did not consider screening as part of disease prevention, and did not consider health screening as a priority in life [32]. Another study reported that low disease perception, limited time, aversion to negative emotion, and previous negative experiences were reasons that the public avoids health screening [33]. 

In this study, 328 subjects (153 men and 175 women) attended the follow-up, and their osteoporosis knowledge scores increased significantly compared to baseline levels. Similar observations were observed in a study that involved men and women over the age of 62 years old (*n* = 376) in a class [34]. Another study demonstrated that lectures and hands-on activities also improved osteoporosis knowledge among 153 young adults aged 18–23 years [35]. Moreover, educational programs developed with theoretical backgrounds increased osteoporosis knowledge among men and women aged 50 and over from three South Florida districts (*n* = 100) [36]. The approach of this study is unique compared to the other studies because education and consultation were personalised and tailored to the bone health condition of the subjects.

Surprisingly, subjects’ perceived susceptibility towards osteoporosis decreased during follow-up. We postulate that subjects remembered their bone health status after the first screening; thus, the majority of them who did not have osteoporosis perceived lower susceptibility towards this condition. On the other hand, their perception of the seriousness of osteoporosis increased during follow up. Moreover, subjects also had a significantly higher perception of the benefits of exercise and calcium intake during follow-up. This observation is within expectations because the importance of exercise and calcium-rich foods was emphasised during the post-screening consultation session. A study on a single-session bone health intervention on Chinese Immigrants in Santa Clara increased calcium intake self-efficacy after two weeks [37]. The lecture or hand-on activities established by Evenson and Sanders (2016) also increased health beliefs regarding exercise and calcium among young adults [35]. Our study showed that even though the subjects were not followed up periodically, the perception of the importance of exercise and calcium-rich foods can be maintained for up to 6 months. 

A significant increase in the consumption of calcium supplements and a reduction in coffee intake were observed during the follow-up. A meta-analysis by Gaines and Marx (2011) reported that educational interventions increased the initiation of calcium supplementation among older men [34]. Another study among Korean elderly people (*n* = 199, aged 50 years or more) reported a decreased percentage of subjects with suboptimal calcium and vitamin D intake after bone health education [38]. However, several obstacles to initiating calcium supplementation have been reported among the subjects, including the lack of need to take up calcium supplements and preference to get calcium from natural food sources. For dairy products, the subjects reported they did not have the habit of consuming them and did not like their taste. This observation agrees with a previous study that reported that Malaysians rarely consume dairy products because they were uncomfortable with the taste of milk [39]. Some subjects reported that they did not take up exercise because they were busy with work and house chores. A study among Malaysians aged 18–55 years also reported that the reasons for the lack of exercise were tiredness after work, laziness, lack of discipline, and family commitment [40]. 

In addition, the BMD of the hip decreased significantly over 6 months. Monitoring of BMD changes should be performed on the same machine and preferably by the same technician as variation will occur on repeat measurements. The error is measured as the coefficient of variance [41] and expressed as a percentage [42]. The DXA machine used in this study has a CV of 1.8% for the spine and a CV of 1.2% for the hip [30]. During the follow-up, BMD changes are considered significant if they exceed the CV values by two times [43]. Therefore, the changes in BMD observed in this study could be due to random error despite being statistically significant. Women may lose up to 20% bone mass within 5–7 years after menopause, followed by a gradual loss at the rate of 0.5–1% annually. For men, bone mass loss occurs with age, but loss begins later in life and continues at about 0.5–1% annually [44]. In another longitudinal study involving men and women aged 60 years and above (*n* = 769) with a follow-up period of 2.5 years, the estimated annual BMD loss was 0.82% annually for men and 0.96% annually for women at the femoral neck [45]. The decline was gradual among these subjects because they had entered a gradual phase of bone loss due to age. The overall rate of decline for spine BMD was −0.17% over 6 months, and that for hip BMD was −1.76% over 6 months in our study. However, long-term changes in BMD would need at least 2 years of data to estimate [46,47]. We also cannot exclude the presence of various physiological, pathophysiologic, anatomical, technical factors and artifacts that could affect BMD readings [48]. Currently, there are no published data on the longitudinal BMD change among Malaysians. 

In this study, 43 new cases of osteoporosis were found. However, out of 43 cases, only 18 patients met doctors with referral letters provided. The percentage of successful referrals (41.86%) was moderate. The main reasons for the subjects asking for a referral letter but not meeting with a physician are due to being busy with work, preferring lifestyle changes at home, or being afraid to take medicines. Most of the subjects with referral letters who went to see physicians were prescribed alendronate plus vitamin D3, while others were prescribed calcium supplements or non-drug lifestyle changes. All subjects were compliant with the prescribed treatment. Many large clinical trials showed that alendronate is effective in increasing BMD, reducing hip and spine fracture risk by half in the first 12–18 months, and improving fracture outcomes. Alendronate is also effective in preventing bone loss in early menopausal women [49]. Meta-analysis studies have also shown that alendronate is the most cost-effective type of treatment in women with low BMD without previous fracture [49]. Apart from alendronate, risedronate, zoledronate, or denosumab could be considered for postmenopausal women without prior fragility fractures or with moderate fracture risk. Raloxifene and ibandronate could be considered alternatives [50]. Calcium and vitamin D supplements are commonly used to prevent osteoporosis. Multiple meta-analyses showed that the combination of calcium and vitamin D increases BMD and reduces fracture risk [51,52]. Other meta-analyses show that vitamin D supplements alone did not exert clinically significant benefits on bone [53,54]. 

The current study does not escape from limitations. Selection bias could not be avoided in this study, as the subjects who volunteered could be more health-conscious, better educated, and in a higher income bracket. The sample size and follow-up period were limited by the constraints on resources. We also did not refer patients with osteopenia to any healthcare providers, which could be a missed prevention opportunity. However, consultation to prevent further bone loss was provided to them. Periodic enforcement of bone health education, which is expected to facilitate the retention of knowledge and improve attitudes and osteoprotective behaviours, was not implemented throughout the six months. Other educational approaches, such as exercise demonstration by physical trainers [38] and sharing sessions by patients living with osteoporosis [13], which were performed in other studies to enhance the belief and behaviour modification of the subjects, were not adopted in the current study. Nevertheless, our approach, which encompasses both bone health screening, personal consultation, and education, encouraged subjects to initiate steps to prevent osteoporosis. Bone health screening has not been routinely used in previous studies [10,36,55], but it is critical to enable subjects to understand their osteoporosis risk and motivate them to make changes for the betterment of their bone health. 

## 5. Conclusions

The current study demonstrated a significant increase in knowledge regarding osteoporosis among Malaysians aged 40 years and above after bone health screening and intervention. The perception regarding susceptibility to osteoporosis decreased, and the perception that osteoporosis is a serious disease and the benefits of exercising increased during the follow-up. For health-related practices, a significant increase in the daily intake of supplemental and dietary calcium and a reduction in coffee or tea drinking were noted during follow-up. The subjects’ barriers to achieving optimal bone health status may be due to the lack of time and awareness of the importance of osteoporosis prevention through diet and lifestyle practices. The percentage of successful referrals was moderate, at only around 41.86%. Being busy at work, favouring lifestyle changes at home, or fearing to take medicine are the reasons for refusing to meet with doctors. For subjects who met their doctors, most of them were given alendronate plus vitamin D3, and the rest were prescribed calcium supplements or lifestyle changes. We recommend regular educational reinforcement and a longer period of follow-up to enhance the knowledge, attitudes, and osteoprotective behaviours of participants and the effects on their BMD. 

## Figures and Tables

**Table 1 ijerph-19-06072-t001:** Characteristics of subjects in this study during the recruitment phase.

Variable of Interest	Mean (SD)			
	Men (*n* = 190)	Women (*n* = 210)	Overall (*n* = 400)	*p*-Value *
**Age (years)**	57.78 (9.58)	56.07 (8.10)	56.88 (8.87)	0.054
**Age of menarche (years)**	-	13.05 (1.87)	-	-
**Number of children (*n*)**	-	2.47 (1.52)	-	-
**Age of menopause (years)**	-	51.08 (3.59), *n* = 146	-	-
**Years since menopause (years)**	-	9.01 (5.98), *n* = 146	-	-
**Body anthropometry**				
**Height (cm)**	167.14 (6.02)	154.51 (5.35)	160.51 (8.49)	<0.001 ^a^
**Weight (kg)**	70.77 (11.59)	60.12 (11.91)	65.18 (12.89)	<0.001 ^a^
**BMI (kg/m^2^)**	25.33 (4.96)	25.22 (4.96)	25.27 (4.48)	0.816
**Body fat percentage (%)**	29.55 (4.92)	40.09 (5.36)	35.08 (7.36)	<0.001 ^a^
**Lean body mass**	47.02 (6.21)	33.60 (5.11)	39.98 (8.78)	<0.001 ^a^
**Waist circumference (cm)**	88.60 (12.38)	82.17 (10.53)	85.22 (11.87)	<0.001 ^a^
**Hip T-score**	−0.61 (1.23)	−1.13 (1.27)	−0.88 (1.28)	<0.001 ^a^
**Spine T-score**	0.17 (1.23)	−0.80 (1.41)	−0.34 (1.41)	<0.001 ^a^
**Hip BMD (g/cm^2^)**	0.93 (0.13)	0.83 (0.12)	0.88 (0.14)	<0.001 ^a^
**Spine BMD (g/cm^2^)**	1.00 (0.16)	0.90 (0.16)	0.95 (0.17)	<0.001 ^a^
** Dietary intake **				
**Energy level (kcal)**	1709.35 (494.45)	1464.03 (457.46)	1581.08 (490.25)	<0.001 ^a^
**Protein (g)**	78.26 (23.78)	69.25 (23.64)	73.52 (24.10)	<0.001 ^a^
**Carbohydrate (g)**	221.23 (71.36)	183.47 (56.71)	201.40 (66.73)	<0.001 ^a^
**Total fat (g)**	60.50 (25.69)	53.42 (28.16)	56.78 (27.21)	0.009 ^a^
**Vitamin A (RE)**	929.11 (611.22)	795.38 (416.67)	858.90 (521.90)	0.010 ^a^
**Sodium (mg)**	3773.68 (1429.63)	3393.68 (1448.00)	3574.18 (1450.00)	0.009 ^a^
**Selenium (ug)**	58.53 (40.01)	49.28 (34.47)	53.67 (37.44)	0.016 ^a^
	*n* (%)			
**Age range**				
**Middle age (40–59 years old)**	100 (52.6)	132 (62.9)	232 (58.0)	0.039 ^b^
**Elderly (60 years old and above)**	90 (47.4)	78 (37.1)	168 (42.0)	
**Ethnicity**				
**Malay**	91 (47.9)	102 (48.6)	193 (48.3)	0.615
**Chinese**	79 (41.6)	90 (42.9)	169 (42.3)	
**Indian**	20 (10.5)	18 (8.6)	38 (9.5)	
**District**				
**Klang**	6 (3.2)	9 (4.3)	15 (3.8)	0.064
**Hulu Langat**	149 (78.4)	178 (84.8)	327 (81.8)	
**Petaling**	23 (12.1)	15 (7.1)	38 (9.5)	
**Gombak**	12 (6.3)	8 (3.8)	20 (5.0)	
**Marital status**				
**Single**	9 (4.7)	20 (9.5)	29 (7.2)	0.065
**Married**	181 (95.3)	190 (90.5)	371 (92.8)	
**Nature of job**				
**Manual**	18 (9.5)	10 (4.8)	28 (7.0)	0.065
**Sedentary**	172 (90.5)	200 (95.2)	372 (93.0)	
**Classification of monthly incomes**				
**B40**	173 (91.1)	206 (98.1)	379 (94.8)	0.002 ^b^
**M40**	17 (8.9)	4 (1.9)	21 (5.3)	
**Highest education level**				
**No formal education**	1 (0.5)	2 (1.0)	3 (0.8)	0.493
**Primary school**	19 (10.0)	14 (6.7)	33 (8.3)	
**Secondary school**	85 (44.7)	112 (53.3)	197 (49.3)	
**Certificate/diploma**	46 (24.2)	46 (21.9)	92 (23.0)	
**University degree**	23 (12.1)	24 (11.4)	47 (11.8)	
**Postgraduate**	16 (8.4)	12 (5.7)	28 (7.0)	
**Current menstrual status**				
**Pre-menopause**	-	41 (19.5)	-	-
**Peri-menopause**	-	23 (11.0)	-	-
**Postmenopause**	-	146 (69.5)	-	-
**Number of lifetime pregnancies (parity)**				
**Nulliparous**	-	36 (17.1)	-	-
**1–3 Pregnancies**	-	104 (49.5)	-	-
**More than 3 Pregnancies**	-	70 (33.3)	-	-
**Dairy intake**				
**Do not drink**	137 (72.1)	113 (53.8)	250 (62.5)	<0.001 ^b^
**Regular drinker**	53 (27.9)	97 (46.2)	150 (37.5)	
**Calcium supplement intake**				
**Yes**	17 (8.9)	38 (18.1)	55 (13.8)	0.008 ^b^
**No**	173 (91.1)	172 (81.9)	345 (86.3)	
**Coffee or tea intake**				
**Do not drink**	30 (15.8)	53 (25.2)	83 (20.8)	0.020 ^b^
**Regular drinker**	160 (84.2)	157 (74.8)	317 (79.3)	
**Alcohol drinking**				
**Non drinker**	125 (65.8)	173 (82.4)	298 (74.5)	<0.001 ^b^
**Ever-drinker**	65 (34.2)	37 (17.6)	102 (25.5)	
**Smoking status**				
**Non-smoker**	112 (58.9)	203 (96.7)	315 (78.8)	<0.001 ^b^
**Ever-smoker**	78 (41.1)	7 (3.3)	85 (21.3)	
**Physical activity status**				
**Inactive**	80 (42.1)	99 (47.1)	179 (44.8)	0.094
**Minimally active**	73 (38.4)	85 (40.5)	158 (39.5)	
**HEPA active**	37 (19.5)	26 (12.4)	63 (15.8)	
**Body mass index**				
**Normal**	16 (8.4)	24 (11.4)	40 (10.0)	0.451
**Underweight**	88 (46.3)	95 (45.2)	183 (45.8)	
**Overweight**	86 (45.3)	91 (43.3)	177 (44.3)	
**Bone health status**				
**Normal**	111 (58.4)	71 (33.8)	182 (45.5)	<0.001 ^b^
**Osteopenia**	68 (35.8)	101 (48.1)	169 (42.3)	
**Osteoporosis**	11 (5.8)	38 (18.1)	49 (12.3)	

SD: standard deviation; * the *p*-values refer to the comparison between men and women; ^a^: indicates a significant difference of *p* < 0.05 as assessed using independent *t*-test; ^b^: indicates a significant difference of *p* < 0.05 as assessed using Chi-square test; B40, subjects with household income < RM 7640; M40, subjects with household income RM 7640–15,159; T20, subjects with household income > RM 15,160.

**Table 2 ijerph-19-06072-t002:** Characteristics of subjects compliant with or lost to follow-up.

Variable of Interest	Mean (SD)		
	Came for Follow Up (*n* = 328)	Lost to Follow Up (*n* = 72)	*p*-Value
**Age (years)**	57.58 (8.58)	53.72 (9.51)	0.002 *
**Body anthropometry**			
**Height (cm)**	160.60 (8.44)	160.10 (8.73)	0.649
**Weight (kg)**	64.22 (12.68)	69.55 (13.02)	0.002 *
**BMI (kg/m^2^)**	24.85 (4.26)	27.20 (4.96)	<0.001 *
**Body fat percentage (%)**	34.88 (7.11)	36.00 (8.42)	0.299
**Lean body mass**	39.57 (8.74)	41.84 (8.75)	0.635
**Waist circumference (cm)**	84.36 (11.82)	89.15 (11.39)	0.002 *
**Hip T-score**	−0.94 (1.25)	−0.60 (1.35)	0.036 *
**Spine T-score**	−0.40 (1.39)	−0.04 (1.46)	0.048 *
**Hip BMD (g/cm^2^)**	0.91 (0.14)	0.87 (0.13)	0.041 *
**Spine BMD (g/cm^2^)**	0.98 (0.17)	0.94 (0.16)	0.065
**Dietary intake (** **only significant results are shown)**			
**Carbohydrate (g)**	197.75 (66.69)	218.56 (64.73)	0.017 *
	Mean % (SD)		
**Knowledge regarding osteoporosis**			
**General knowledge regarding osteoporosis**	70.68 (18.43)	73.15 (14.95)	0.289
**Prevention knowledge regarding osteoporosis**	63.41 (16.83)	62.73 (16.67)	0.755
**Total knowledge regarding osteoporosis**	67.05 (13.27)	67.94 (13.02)	0.605
**Beliefs regarding osteoporosis**			
**I: Perceived susceptibility to osteoporosis**	59.76 (13.90)	57.22 (13.96)	0.162
**II: Perceived seriousness of osteoporosis**	72.68 (18.58)	69.44 (20.41)	0.189
**III: Perceived benefits of exercise**	80.37 (12.01)	78.33 (14.54)	0.271
**IV: Perceived benefits of calcium intake**	77.87 (12.50)	80.00 (12.10)	0.187
**V: Barriers to exercise**	51.40 (15.44)	53.06 (15.89)	0.414
**VI: Barriers to calcium intake**	44.79 (9.58)	44.31 (10.85)	0.707
**VII: Health motivation**	74.88 (10.39)	73.43 (10.58)	0.285
**Total beliefs regarding osteoporosis**	63.95 (5.65)	63.10 (5.92)	0.252
**Dairy intake**			
**Do not drink**	201 (61.6)	49 (68.1)	0.282
**Regular drinker**	127 (38.7)	23 (31.9)	
**Calcium supplement intake**			
**Yes**	46 (14.0)	9 (12.5)	0.734
**No**	282 (86.0)	63 (87.5)	
**Coffee or tea intake**			
**Do not drink**	72 (22.0)	11 (15.3)	0.206
**Regular drinker**	256 (78.0)	61 (84.7)	
**Alcohol drinking**			
**Non drinker**	239 (72.9)	59 (81.9)	0.109
**Ever-drinker**	89 (27.1)	13 (18.1)	
**Smoking status**			
**Non-smoker**	264 (80.5)	51 (70.8)	0.070
**Ever-smoker**	64 (19.5)	21 (29.2)	
**Physical activity status**			
**Inactive**	142 (43.3)	37 (51.4)	0.426
**Minimally active**	132 (40.2)	26 (36.1)	
**HEPA active**	54 (16.5)	9 (12.5)	

SD: standard deviation. * indicates a significant difference between the two groups.

**Table 3 ijerph-19-06072-t003:** Characteristics of subjects before and after the intervention.

Variable of Interest	Mean (SD)		
	Baseline (*n* = 328)	Follow-Up (*n* = 328)	*p*-Value
**Age (years)**	57.60 (8.57)	57.60 (8.57)	1.000
**Body anthropometry**			
**Height (cm)**	160.59 (8.44)	160.59 (8.44)	1.000
**Weight (kg)**	64.24 (12.73)	64.17 (13.35)	0.119
**BMI (kg/m^2^)**	24.86 (4.29)	24.35 (4.88)	0.420
**Body fat percentage (%)**	34.89 (7.11)	34.88 (6.95)	0.686
**Lean body mass**	39.57 (8.74)	40.25 (8.75)	0.610
**Waist circumference (cm)**	84.39 (11.89)	86.49 (11.13)	0.382
**Hip T-score**	−0.94 (1.25)	−1.06 (1.22)	<0.001 *
**Spine T-score**	−0.40 (1.39)	−0.40 (1.41)	0.971
**Dietary intake (only significant results are shown)**			
**Calcium (mg)**	604.79 (20.54)	644.90 (23.26)	0.018 *
**Copper (mg)**	0.88 (0.06)	1.08 (0.11)	0.042 *
**Selenium** ** (mg) **	55.08 (2.18)	52.86 (2.21)	0.024 *
**α-tocopherol (mg)**	17.03 (7.61)	15.26 (7.06)	0.028 *
	Mean % (SD)		
**Knowledge regarding osteoporosis**			
**General knowledge regarding osteoporosis**	70.58 (18.45)	78.25 (17.13)	<0.001 *
**Prevention knowledge regarding osteoporosis**	63.36 (16.80)	74.64 (15.58)	<0.001 *
**Total knowledge regarding osteoporosis**	66.97 (13.26)	76.45 (12.32)	<0.001 *
**Beliefs regarding osteoporosis**			
**I: Perceived susceptibility to osteoporosis**	59.76 (13.90)	56.55 (13.61)	<0.001 *
**II: Perceived seriousness of osteoporosis**	72.68 (18.58)	75.55 (19.31)	0.010 *
**III: Perceived benefits of exercise**	80.37 (12.01)	82.20 (11.39)	0.002 *
**IV: Perceived benefits of calcium intake**	77.87 (12.50)	80.49 (10.36)	0.107
**V: Barriers to exercise**	51.34 (15.45)	49.15 (15.00)	0.060
**VI: Barriers to calcium intake**	44.79 (9.58)	44.51 (9.44)	0.613
**VII: Health motivation**	74.88 (10.39)	74.37 (10.56)	0.266
**Total beliefs regarding osteoporosis**	63.94 (5.65)	63.48 (5.30)	0.118
**Dairy intake**			
**Do not drink**	201 (61.3)	202 (61.6)	1.000
**Regular drinker**	127 (38.7)	126 (38.4)	
**Calcium supplement intake**			
**Yes**	46 (14.0)	72 (22.0)	<0.001 *
**No**	282 (86.0)	256 (78.0)	
**Coffee or tea intake**			
**Do not drink**	72 (22.0)	102 (31.1)	<0.001 *
**Regular drinker**	256 (78.0)	226 (68.9)	
**Alcohol drinking**			
**Non drinker**	239 (72.9)	239 (72.9)	1.000
**Ever-drinker**	89 (27.1)	89 (27.1)	
**Smoking status**			
**Non-smoker**	264 (80.5)	264 (80.5)	1.000
**Ever-smoker**	64 (19.5)	64 (19.5)	
**Physical activity status**			
**Inactive**	142 (43.3)	142 (43.3)	1.000
**Minimally active**	132 (40.2)	132 (40.2)	
**HEPA active**	54 (16.5)	54 (16.5)	

* indicates a significant difference between the two groups.

**Table 4 ijerph-19-06072-t004:** Changes in hip and spine BMD after the intervention.

Variables	Category, *n*	Mean (SD), g/cm^2^	*p*-Value
**Overall**			
**Spine BMD**	**First phase (*n* = 328)**	0.94 (0.16)	0.206
	**Follow up phase (*n* = 328)**	0.94 (0.17)	
**% changes**			−0.17 (2.98)
**Hip BMD**	**First phase (*n* = 328)**	0.87 (0.14)	<0.001 *
	**Follow up phase (*n* = 328)**	0.85 (0.14)	
**% changes**			−1.76 (4.38)
**Men**			
**Spine BMD**	**First phase (*n* = 153)**	1.00 (0.16)	0.128
	**Follow up phase (*n* = 153)**	1.00 (0.16)	
% changes			0.41 (2.96)
**Hip BMD**	**First phase (*n* = 153)**	0.93 (0.13)	<0.001 *
	**Follow up phase (*n* = 153)**	0.91 (0.14)	
**% changes**			−1.82 (4.17)
**Women (Pre-menopause)**			
**Spine BMD**	**First phase (*n* = 183)**	1.00 (0.15)	<0.001 *
	**Follow up phase (*n* = 183)**	1.00 (0.15)	
**% changes**			0.36 (2.95)
**Hip BMD**	**First phase (*n* = 183)**	0.92 (0.13)	< 0.001 *
	**Follow up phase (*n* = 183)**	0.90 (0.14)	
**% changes**			−1.78 (4.01)
**Women (Peri-menopause)**			
**Spine BMD**	**First phase (*n* = 183)**	1.02 (0.13)	<0.001 *
	**Follow up phase (*n* = 183)**	1.00 (0.13)	
**% changes**			−1.39 (2.07)
**Hip BMD**	**First phase (*n* = 20)**	0.85 (0.09)	<0.001 *
	**Follow up phase (*n* = 20)**	0.85 (1.00)	
**% changes**			−0.22 (7.23)
**Women (Postmenopause)**			
**Spine BMD**	**First phase (*n* = 125)**	0.85 (0.15)	<0.001 *
	**Follow up phase (*n* = 125)**	0.84 (0.15)	
**% changes**			−0.75 (3.00)
**Hip BMD**	**First phase (*n* = 125)**	0.80 (0.12)	<0.001 *
	**Follow up phase (*n* = 125)**	0.79 (0.12)	
**% changes**			−1.99 (4.30)

CV for spine: 1.8%; CV for hip: 1.2%; * indicates significant difference of *p* < 0.05.

**Table 5 ijerph-19-06072-t005:** Examples of written feedback from the subjects for not meeting medical doctors for further consultation.

Reasons	Example of Response
**Changes of lifestyle at home**	“My nephew taught me to exercise at home”
	“I started to consume calcium supplements and dairy products after consulting with doctor”
**Busy with work/class**	“I need to attend 10 slamic class 5 days a week”
	“I am busy with work and have no time to meet the doctor”
	“Waiting time at hospital was too long, I have no time to wait so long”
**Fear of taking medication**	“I am afraid of taking medication and later kidney failure”
**Medical cost was high**	“I can’t afford high medical costs because I am already retired, with no income”
**Hospital was far from home**	“I went to clinic but was referred to hospital; the hospital was far, so I didn’t go”

## Data Availability

The data are available at reasonable request to the corresponding author.

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
