# Peer review of "Effect of a Screening and Education Programme on Knowledge, Beliefs, and Practices Regarding Osteoporosis among Malaysians"

_ijerph, 2022, doi:10.3390/ijerph19106072_

Round 1
Reviewer 1 Report
The present study undelines
the importance of detecting new cases
of osteoporosis and follow up,
even for a short period of time
Author Response
Thank you for the comments.
Reviewer 2 Report
Osteoporosis is a common public health problem. This study addresses a very important topic - actions to improve health-seeking behaviors to prevent the occurrence of osteoporosis. The study is well designed and described. The conclusions are valid. I have only minor comments regarding the treatment of osteoporosis. Please remove the trade name of the drug (Fosamax Plus) from the abstract and conclusions. Please replace it with the international name - combination of alendronate and vitamin D. Writing the name Fosamax Plus in many places of the manuscript suggests that the study was funded by a pharmaceutical company. Additionally, I propose to remove the statement "Fosamax Plus (a combination of alendronate and vitamin D) is the main treatment for men and women facing osteoporosis problems [48,49]."
Author Response
Dear reviewer,
Thank you for reviewing our manuscript. We appreciate your comments and have responded to them in a point-by-point manner in the attached response sheet.

Reviewer 3 Report
Dear Sir/Madam,
Thank you for the opportunity to review the manuscript “Effect of a screening and education programme on knowledge, beliefs and practices regarding osteoporosis among Malaysians”. The subject of the article is important, as prevention is a key to decrease fracture risk.
I have the following comments on the manuscript:
Material and Methods
Line 95 Please clarify whether the answers on the questionnaire were self-reported and if the questionnaires were filled in at the study centre or at home before the study visit.
Line 105 – 108 Please add brief information on how the scores are calculated and whether higher values represent increased or decreased belief of the participant.
Line 128 – 130 Was the waist circumference measured at two different points, i.e. the lowest rib margin and the iliac crest? Was the waist-hip ratio calculated? If both measurements were made, which level is used for the variable waist circumference in table 1, 2 and 3?
Line 154 Was it the exactly same screening that was performed at the follow-up or were there any changes? Additional examinations or anything removed?
Were osteopenic subjects given any special recommendations or referrals?
Results
Line 184 Do the authors mean menopausal or rather postmenopausal i.e. having passed the menopause previously? Please change to postmenopausal if the authors are referring to women that have passed their menopause previously.
Table 1 Please add an explanation to “p-value” indicating that this p-value represents the difference between men and women in the study, if this is the correct interpretation of the p-value. Otherwise, please indicate what it represents.
“Weight” is missing its measurement unit.
“Menopause” please see comment above, is it post-menopausal that it is referring to?
“HEPA Active”, please clarify in the material and methods section what this means and add a clarification in the footnote of the table 1, 2 and 3.
Line 193 – 195 Did the BMD at baseline in non-attendees differ from the BMD at baseline in the participants that attended their follow-up?
Table 2 As written above, please indicate somewhere in the material and methods section how the scoring systems for “Beliefs and Knowledge regarding osteoporosis” are calculated.
“Weight” is missing its measurement unit.
Table 3 How can the significance for the different groups of “Dairy intake” be of so high level significant when the number of individuals in the groups are identical?
There are no p-values for “Coffee or tea intake”, “Alcohol drinking”, “Smoking status” and “Physical activity status”; is there any particular reason for why they are missing?
In “Physical activity status” the number of HEPA active participants are missing, only the percentage is shown.
Table 5 For “Women (Pre-menopause)” the % change noted as -4.80 (18.25), but the values for first and follow-up phase are identical, is this due to clerical error or due to distribution of data?
Similarly, for “Women (Peri-menopause)”, the largest percentage of change is indicated at -14.70 (37.43), but the values are identical rendering the same question as above.
Line 240 – 242 Please clarify if the new cases of osteoporosis were all participants that came back for their follow-up visit or were any of them in the group that did not come back? Because in table 6 the number of individuals that were referred for follow-up is shown, but the numbers in the column does not add up unless the participants in the row “Did not come back for follow-up” are excluded.
Line 254 Please clarify what is included in the word “treatment”.
Discussion
Line 267 – 268 It is stated that the coffee and tea drinking decreased during follow-up, but in table 3 there is no p-value to support this statement, please see previous comment. The actual number of abstainers from coffee and tea has decreased, but has this been tested with a statistical test or not?
Line 270 – 271 Please clarify that the prescribed treatment may be one or several of the stated measures.
Line 306 Please see comment regarding line 267 – 268.
Line 321 and 322 For the discussion, could it be argued that the overall significant decrease in spine BMD could be due to the large decrease in postmenopausal women? Could this decrease be considered part of the physiological changes in postmenopausal women? Nonetheless, it is quite amazing that such a large decrease in BMD was observed after such a short follow-up period as 6 months.
Please also, consider previous remarks regarding menopausal vs. postmenopausal women and change accordingly.
For information only, there is a very recent review in the subject that could be interesting to the authors: Rubæk M, Hitz MF, Holmberg T, Schønwandt BMT, Andersen S. Effectiveness of patient education for patients with osteoporosis: a systematic review. Osteoporos Int. 2022 May;33(5):959-977. doi: 10.1007/s00198-021-06226-5. Epub 2021 Nov 12. PMID: 34773131.
Author Response

(The authors gave the same response as above.)

Reviewer 4 Report
Thanking you for asking me to review this paper.
This paper looks at the practicalities of bone health screening and education and its effectiveness over a six-month period.
General comments
The design of the paper is somewhat flawed. I understand that the population in Kuala Lumpur is about 7.5million, but they have chosen to in the first phase of the study look at 400 Malaysian men and women. The subjects appeared to be somewhat selected as they appear to be better educated and are in a higher income bracket. Subjects are relatively young. It is also unclear as to why six months was chosen rather than twelve months. They have also excluded people with fractures and those who are unable to be scanned and this needs clarification.
Material and methods: This section is too long and needs to be shortened and needs to be more precise and also needs to include reasons why fractures were excluded.
Statistical analysis seems satisfactory
In the results section the subjects are overall young and of diverse ethnicity. In regards to the tables they are very comprehensive and long and it would be advisable that some of the data in most of the tables was included in the results section as texts, especially table 4 onwards to table 9. I am unsure as to whether table 9 has been correctly labelled.
Discussion: There is mention of the decreased susceptibility to osteoporosis knowledge and lines 301 – 305 there is mention of the benefits of calcium. There is little or no evidence of the evidence of calcium supplements in the age group they have studied and this needs to be mentioned.
In the section regarding bone density even though the bone density was either normal or mildly osteopenic, they said that the bone density decreased over a period of 6 months, although they have alluded to this in the discussion, they have said that these decreases are clinically significant. Having excluded patients with fractures and having no prospective data on fractures and fracture outcomes, its not appropriate to suggest that this bone mineral density decrease is clinically significant.
In lines 350 – 352 they refer to osteoporosis problems. This needs to be reworded and other treatment options need to be discussed. In lines 361 – 371 they have discussed the imitations of their study. In light of my earlier comments about the overall population of Kuala Lumpur and highly selective small group of patients and the short follow up time of six months needs to be expanded on in the area of the discussion.
Author Response

(The authors gave the same response as above.)
